# Dopamine builds and reveals reward-associated latent behavioral attractors

Jérémie Naudé [1,2] ✉, Matthieu X. B. Sarazin[3], Sarah Mondoloni [1], Bernadette Hannesse[1], Eléonore Vicq[4], Fabrice Amegandjin[1], Alexandre Mourot [1,4], Philippe Faure [1,4,5] ✉ & Bruno Delord[3,5] ✉

Phasic variations in dopamine levels are interpreted as a teaching signal reinforcing rewarded behaviors. However, behavior also depends on the motivational, neuromodulatory effect of phasic dopamine. In this study, we reveal a neurodynamical principle that unifies these roles in a recurrent network-based decision architecture embodied through an action-perception loop with the task space, the MAGNet model. Dopamine optogenetic conditioning in mice was accounted for by an embodied network model in which attractors encode internal goals. Dopamine-dependent synaptic plasticity created "latent" attractors, to which dynamics converged, but only locally. Attractor basins were widened by dopamine-modulated synaptic excitability, rendering goals accessible globally, i.e. from distal positions. We validated these predictions optogenetically in mice: dopamine neuromodulation suddenly and specifically attracted animals toward rewarded locations, without off-target motor effects. We thus propose that motivational dopamine reveals dopamine-built attractors representing potential goals in a behavioral landscape.

Transient, phasic dopamine (DA) release contributes both to learning (updating the values of actions, used to make future decisions based on experience) and to motivation (affecting ongoing decisions and invigorating goal-oriented behaviors), but reconciling these two roles within a unified theory of DA function has remained challenging[1,2]. The popular reinforcement learning (RL) theory interprets phasic DA signaling as a reward-related teaching signal[2,3], which functions by modulating long-term synaptic plasticity[4,5] to build neural representations of the value of actions that have previously led to reward[6]. This role of DA in value learning is well demonstrated by the robust conditioned place preference induced by optogenetic stimulation of DA cells in the ventral tegmental area (VTA)[7].

Although the original RL theory did not define a role for phasic DA signaling in ongoing behavior[2,3], there has been renewed interest in both experimentally linking phasic DA activity with motivation[8–11] and building RL models that account for motivational DA. A large body of

evidence supports that phasic DA neuron activity occurs just before self-paced movement initiation[9–13]. However, the causal role of such phasic DA activity in the ongoing movement remains debated. While phasic optogenetic excitation or inhibition of DA neurons have been found to affect action initiation in some studies[8,10,11,14], in other settings manipulating DA activity did not have any effect on ongoing behavior[10,15,16]. These conflicting results have proven hard to reconcile within a RL framework, despite continuous efforts.

Theoretical accounts suggest either a "directional" role with DA signals specifying the decision to be taken[17] or an "activational" (or energizing) role, with DA determining the action latency[14,18] and/or the level of motor resources to engage in performing an action[1,19–21]. The encoding capacity of DA cells is limited[22], which suggests that their role is not directional (but see[17,21]). However, if DA gates decision-making by lowering a decision threshold, increasing the probability and reducing the latency of actions[19,23], it remains unclear how such action gating by

[1]Sorbonne Université, Inserm, CNRS, Neuroscience Paris Seine; Institut de Biologie Paris Seine (NPS - IBPS), Paris, France. [2]INSERM, CNRS, Université de Montpellier; Institut de Génomique Fonctionnelle, Montpellier, France. [3]Institut des Systèmes Intelligents et de Robotique (ISIR), Sorbonne Université, CNRS, Paris, France. [4]Brain Plasticity Laboratory, CNRS UMR 8249, ESPCI Paris; PSL Research University, Paris, France. [5]These authors contributed equally: Philippe Faure, Bruno Delord. ✉e-mail: jeremie.naude@igf.cnrs.fr; phfaure@gmail.com; bruno.delord@gmail.com

DA would also affect the content (i.e. speed or direction) of movements. Moreover, it is important to note that manipulating phasic DA often does not have the same impact on all actions[8,14], which contradicts latency or decision-threshold models that would predict content-independent DA effects. DA signaling is primarily associated with, and necessary for, non-stereotyped, anticipatory, distal, or effortful behaviors, i.e. when some physical or cognitive distance separate the animal from a reward[8,24,25]. Therefore, the reason why manipulating DA activity can affect both action latency, action direction, and movement vigor, but only in certain animal states and behavioral settings, remains totally enigmatic.

In the following, rather than starting from behavioral DA effects to build a phenomenological model, we sought to express DA motivational roles as a dynamical consequence of the known biophysical effects of phasic DA on target neural networks. DA modulation of synaptic plasticity is believed to build "Hebbian" assemblies[26] of strongly interconnected neurons, representing a decision that was repeatedly rewarded. Activity within such Hebbian assemblies constitute goal-encoding attractors[27,28], which attract network dynamics in their vicinity (i.e. their basin of attraction in the state-space). In standard attractor models, convergence from an arbitrary current state to a goal-encoding attractor can be either triggered by a specific cue stimulus[27] or driven by noise fluctuations[28]. However, humans and other animals can self-initiate goal-directed movements. Cue-induced convergence to a goal-encoding attractor does not account for instrumental decisions that are autonomously generated, based on internal (action-outcome) associations[29]. Noise-induced switching to a goal-encoding attractor also provides an unsatisfactory account for internally-generated decisions that are driven by motivational states, rather than simply occurring randomly[30]. We therefore assessed the possibility that phasic DA, based on its biophysical action, could provide a motivational signal affecting goal-encoding attractors online, i.e. during the decision process itself.

Here, by testing dynamical model predictions with experimental data, we demonstrate that the motivational role of phasic DA signaling is to reveal latent network attractors previously built by DA-modulated plasticity, thereby promoting the engagement of network activity into decision-related attractor dynamics. Specifically, we present a recurrent network-based decision architecture hereafter referred to as "Motivational Attraction to Goals by Network dynamics" ("MAGNet") model. MAGNet is embodied, through an action-perception loop, within the task space. We demonstrate how DA revealing latent – i.e. not systematically expressed – attractors generates goal-directed actions toward previously rewarded locations. Therefore, we reinterpret the motivational role of phasic DA signaling as controlling the accessibility of attractors representing behavioral goals within a behavioral energy landscape.

## Results

### Optogenetics stimulation of VTA DA cells produces precise place preference and motivated behaviors

To characterize the role of phasic, transient dopamine (DA) signaling from the ventral tegmental area (VTA) in both reinforcement learning and motivation, we used an un-cued optogenetic conditioning task. Indeed, cues paired with reward induce both the release of DA and an increase in motor responses[31], confounding the interpretation of the role of DA in subsequent behavior. Instead, we designed a task which requires mice to learn an internal memory of rewarded locations. We achieved selective manipulation of dopamine neurons by selectively expressing channelrhodopsin (ChR2) in the VTA dopamine neurons from dopamine transporter (DAT)-Cre mice (Fig. 1a). In a circular open-field, DA neurons were stimulated (500 ms, 20 Hz VTA photostimulation, Fig. 1a, which drives bursting activity in dopamine neurons, Supp Fig. 1) when mice were detected in one among three explicit locations. Mice cannot receive two consecutive photostimulations on the same location, so they alternated between the rewarded locations[13,32,33]. Mice increased the number of photostimulations earned with learning sessions (Fig. 1b). This increased performance in ChR2 mice was due to a decrease in the distance traveled between successive rewarded locations, compared to controls (Fig. 1c, top), together with an increase in maximal speed (Fig. 1c, bottom). Hence, increased performance following place-

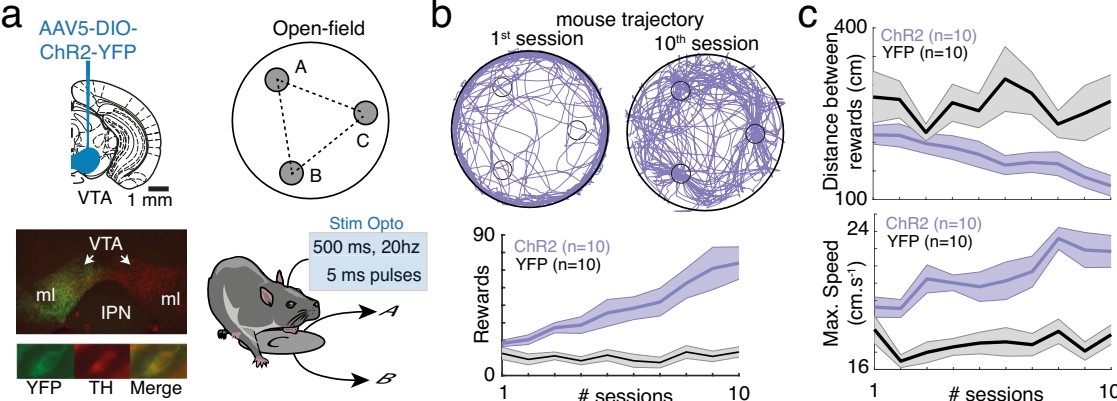

**Fig. 1 | Sequential place preference using phasic VTA DA photostimulation interrogates DA roles in reinforcement and motivation. a** Left, double-floxed inverted open-reading frame (DIO) channelrhodopsin 2 - yellow fluorescent protein (ChR2-YFP)-expressing adeno-associated virus (AAV5) was injected in ventral tegmental area (VTA) from dopamine transporter (DAT)-Cre mice, resulting in selective expression of ChR2-YFP in tyrosine hydroxylase (TH)-expressing dopamine (DA) neurons. IPN: interpeduncular nucleus, ml : medial lemniscus. Right, VTA photostimulation was delivered when mice were detected within one of three explicit locations (A, B, C) in the open field. Mice could not receive two consecutive photostimulations at the same location (e.g. C), so they alternated between locations (e.g. A or B after C). **b** Top, trajectories (10 min) of one mouse expressing ChR2 in the VTA (purple) at the beginning (left) and at the end (right) of the learning sessions. Bottom, number of photostimulations against session number for Chr2-expressing (purple) and YFP-expressing (black) animals. Two-way ANOVA with repeated measures, groups: $F_{(1)}$ = 30.04, $p$ = 3.10^{-5}, time: $F_{(9)}$ = 5.69, $p$ = 7.10^{-7}, interaction: $F_{(1,9)}$=3.9, $p$ = 2.10^{-4}. **c** Distance between two consecutive reward locations (top, two-way ANOVA with repeated measures, groups: $F_{(1)}$=50.53, $p$ = 1.10^{-6}, time: $F_{(9)}$=7.23, $p$ = 8.10^{-9}, interaction: $F_{(1,9)}$=5.43, $p$ = 2.10^{-6}) and maximal speed (bottom, two-way ANOVA with repeated measures, groups: $F_{(1)}$=71.65, $p$ = 1.10^{-7}, time: $F_{(9)}$=8.84, $p$ = 8.10^{-11}, interaction: $F_{(1,9)}$=2.18, $p$ = 0.03) against session number for Chr2-expressing (purple) and YFP-expressing (black) animals. Data are presented as mean ± s.e.m. See also source data file.

photostimulation pairings was due to a combination of directional and activational effects, which characterize motivated behaviors[1,20]. In the following, we sought to systematically dissect how such increases depend on the different roles proposed for VTA DA.

## Goal-directed actions in an embodied biophysical recurrent neural network

To dissect the roles of DA in reinforcement, through DA effects on synaptic plasticity (DA-plasticity), and in motivation, through online DA biophysical effects (DA-excitability), in our conditioning task (Fig. 1), we developed a biophysical model consisting of a decision architecture assessing how an artificial mouse (i.e. an "e-mouse") navigates under DA regulation (Fig. 2a). The model is referred to as the 'Motivational Attraction to Goals by Network Dynamics' (MAGNet) hereafter. As place-reward association relies on a distributed circuit comprising the PFC, basal ganglia, thalamus, hippocampus and amygdala[34], we designed MAGNet as a distributed decision architecture, with different degrees of biological realism. To assess the impact of DA on attractor dynamics, goals were encoded by a recurrent neural network model of leaky integrate-and-fire neurons, with detailed biophysical realism (Fig. 2a, blue, Methods). This recurrent network can be considered as the prefrontal stage of decision-making (although only attractor dynamics, but not the prefrontal location, is mandatory for the following results, see Discussion). This model network displayed mixed selectivity, i.e. neuronal encoding of both space (the current and desired animal positions) and reward (through DA-mediated learning). The network was organized topologically: neurons had a receptive field for the mouse position, i.e. feed-forward inputs putatively from hippocampal place cells[34,35] and in turn biased the animal's goal toward their preferred location (i.e. the same location as their receptive field, Fig. 2a, black). To restrain the model dimensions, and ensure that the effects are primarily due to attractor dynamics, we modeled the other stages of decision-making algorithmically. The internal goal was decoded from the recurrent network using a softmax selection rule, potentially representing some of the basal ganglia operations (Fig. 2a, orange). Finally, the e-mouse converged toward its internal goal with speed ballistics, accounting for commands set by motor structures (Fig. 2a, brown).

The recurrent neural network was thus embodied, in the sense that its activity determined the navigation of the e-mouse, which subsequently affected the spatial feedback input to the network. Hence, this formed an action-perception loop with the environment. Operation of such an embodied decision architecture fundamentally differs from that of a simple input-output network architecture. This is due to the non-trivial, circular causality between the animal and the task space it is immersed in. When spiking in the recurrent network was dominated by inputs encoding the e-mouse position (black dot) (gray squares, Fig. 2b, upper maps), the internal goal (orange dot) was determined by the e-mouse position. When the internal goal was confounded with the current position, a default behavior (i.e. circling along walls with some inroads, see below and *Methods*) mimicked that of real mice before learning. Conversely, a significant bump of activity in neurons encoding for a position distant from the current e-mouse position (as artificially introduced for illustration purposes in Fig. 2b lower map, green dot) resulted in a shift of the internal goal to that distant position. Consequently, navigation was dominated by a convergence to the position encoded by the bump, i.e. the e-mouse position was driven toward the internal goal (Fig. 2b, green arrows). Therefore, in the embodied decision network, goal-directedness can be formalized as a switch from input-driven to internally-generated dynamics (i.e. an attracting activity bump at the rewarded location). As we shall see below, this framework allows us to assess the role of motivational dopamine in terms of recurrent network dynamics.

## Motivation emerges through two distinct biophysical effects of dopamine

We considered two different effects of DA on the biophysical properties of the neurons in MAGNet. DA enables long-term synaptic plasticity in cortical/subcortical areas[4–6]. By reinforcing synaptic weights, DA links sensory states to rewarded actions[3]. Here, we modeled DA consolidation of spike-timing dependent plasticity (STDP): correlated pre-postsynaptic activity led to eligibility traces (or synaptic tags, Fig. 2c, light blue) that were transformed by DA into actual excitatory synaptic changes[4–6]. However, DA also modulates effective synaptic excitability by instantaneously potentiating the efficacy of NMDA currents, which are paramount in setting network dynamics[36,37]. To disentangle the behavioral effects associated with these two biophysical properties, we considered different versions of the model including DA consolidation of synaptic plasticity (DA-plasticity; green arrows), instantaneous DA NMDA upregulation (DA-excitability; red arrow), or both (DA-plasticity-excitability; Methods). Simulated phasic DA was delivered as a reward when the e-mouse crossed the rewarded locations, but also randomly during navigation to account for spontaneous DA occurring in mice (9; see Methods and below). Prior to learning, navigation was governed by default behavior toward and along arena walls (with some incursion into the arena; Methods, Fig. 2d left). As observed with real mice (Fig. 1b), e-mice learned three place–reward associations when navigating in the arena (Fig. 2d right). Both DA effects on plasticity and excitability amplified the directional (decreased distance to reward, Fig. 2e) and activational (increased maximum speed, Fig. 2f) effects of DA, resulting in increased performance (Fig. 2g), as in real mice (Fig. 1b,c). The symmetric nature of the graphs (Fig. 2e–g) suggests a synergistic effect of the two properties. Hence, multiple combinations of increases in both DA-excitability and DA-plasticity could equally account for our experimental data, suggesting, in turn, that RL-type explanations of decision-making exclusively based on DA-plasticity may be incomplete. We next assessed whether fluctuations in the phasic DA activity occurring before mice started a navigation bout towards the reward location (hereafter "pre-movement" DA activity), which is observed in similar settings[9–13], may help distinguish between the effects of DA-plasticity+excitability and DA-plasticity alone. We considered two alternative scenarios. First, pre-movement DA activity could passively reflect the history of previous DA release (i.e., coding the magnitude of DA-plasticity, Fig. 2h), as would be observed with a reward prediction error containing a prediction term (e.g. a time-difference RPE[3,10]). Second, pre-movement DA activity could constitute a motivational command on ongoing behavior (i.e., coding the magnitude of DA-excitability, Fig. 2i). Whatever the scenarios, DA activity correlated with movement speed, further suggesting that standard experimental measures cannot distinguish between DA-plasticity+excitability and DA-plasticity alone.

We then implemented the MAGNet model with a single rewarded location in the center of the arena to (i) better distinguish between long-term (DA plasticity) and on-line (DA excitability) effects of DA signaling on decision making. With DA-plasticity only (Fig. 3b), simulated phasic DA (see Methods) delivered when the e-mouse crossed the rewarded location (which occurred by chance in naïve e-mice) yielded long-term synaptic plastic modifications (top panels) that accumulated over trials (bottom). The resulting strongly-connected Hebbian assembly encoded the place–reward association (right panel). By contrast, DA-excitability only transiently increases synaptic efficacy (<1 s;[24]) in the whole network, as a consequence of NMDA potentiation on a short timescale (Fig. 3c).

When navigating in the arena, the e-mouse converged more toward the rewarded location when considering that DA affected

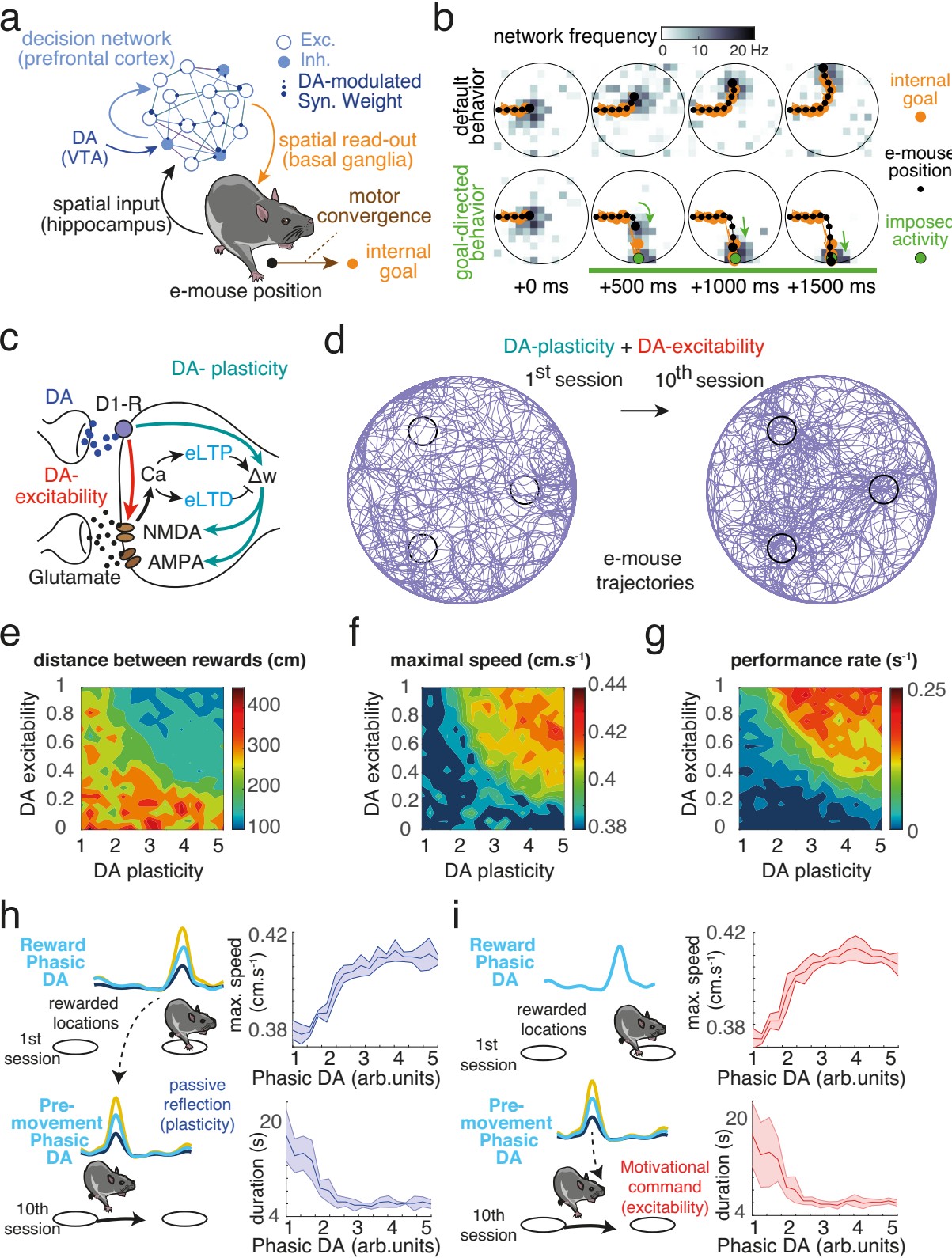

both plasticity and excitability (Fig. 3d, center, bold trajectories), rather than only plasticity or only excitability (Fig. 3d, left and right). In the DA-plasticity+excitability condition, instantaneous NMDA potentiation (i.e. DA-excitability) had a larger, multiplicative effect on synapses already potentiated by DA-plasticity, resulting in a massive co-activation of neurons from the Hebbian assembly (Fig. 3e). DA-plasticity+excitability thus set the internal

goal (orange trajectory) on the learned reward location, attracting the e-mouse (black trajectory).

## Reduced theoretical model uncovers that dopamine reveals latent attractors

We then exploited the radial symmetry of the environment to provide a reduced equation accounting for DA effects on animal behavior, and to

**Fig. 2 | Behavioral performance and correlations with dopaminergic activity do not disentangle reinforcing and motivational roles of dopamine. a** The decision architecture of MAGNet comprises a biophysical prefrontal (PFC) recurrent network of excitatory (exc.) and inhibitory (inh.) neurons, with DA-modulated excitatory synapses, hippocampal position-encoding inputs (black), basal ganglia internal goal soft-max-decoding (orange) and motor convergence toward the internal goal (brown). **b** hippocampal inputs impose an activity bump (dark gray in maps). Under default behavior (upper maps), the e-mouse position (black dot) and internal goal (orange dot) are conjoined, such that goal-directed behavior is inoperant and navigation oriented toward the wall. Under goal-directed behavior (lower maps), the internal goal is decoded at a larger, distant, activity peak (artificially created here, green dot) such that the e-mouse converges to the goal (arrows). **c** Modeling DA biophysics in MAGNet. At excitatory synapses, DA, through D1 type receptors (D1-R), consolidates calcium-induced early eligibility traces (eLTD for long-term depression and eLTP for long-term potentiation) into long-term weight changes affecting excitatory (AMPA and NMDA) currents (DA-plasticity; green arrows). DA also instantaneously upregulates NMDA maximal conductances (DA-excitability; red arrow). **d** e-mouse trajectories during the 1st and 10th session of simulated protocol where DA-plasticity and DA-excitability operated online. **e–g** Maps represent the average distance (e) and the maximum speed (f) between consecutive reward locations and the reward rate (g), as a function of DA-plasticity (maximal weights of Hebbian Assemblies) and DA-excitability (NMDA scaling factor). **h,i** The magnitude of phasic DA classically observed at movement onset may (**h**, left) passively reflect the extent of previous DA-modulated plasticity or (**i**, left) constitute an active command affecting ongoing behavior through excitability effects. Correlations between the magnitude of phasic DA before the movement and maximal speed (top right) or trial duration (bottom right) can be observed (indiscriminately) in either scenarios. Data are presented as mean ± s.e.m.

derive experimental predictions. According to these biophysically-informed e-mouse simulations, instantaneous DA-excitability reveals long-term DA-plasticity reinforcement and drives goal-directed actions in mice. We analytically derived from the biophysical model a reduced model, which captures the hypothesized dynamical effects of DA without having the large number of free parameters of the biophysical model (Methods). In the MAGNet theory, the decision architecture including the animal position, neural network activity and internal goal, can be captured through a one-dimensional behavioral potential energy (BPE) governing e-mice behavior, similar to a particle in an energy landscape (see methods). Because of the revolution symmetry of the one-reward environment, BPE could here be determined as

$$E_P^{behavior}(p, DA) = \alpha_w w(p) \hat{I}_{Exc}(DA) + \frac{1}{2}\alpha_g \rho DA p^2 \quad (1)$$

with $p$ the e-mouse distance to the rewarded Hebbian assembly location ($P_{HA} = 0$), $\hat{I}_{Exc}$ the DA-dependant average inward recurrent network excitatory current (per weight unit, $w(p)$) DA-reinforced synaptic incoming weights' sum (displaying a uniform vs gaussian spatial distribution before vs after learning, respectively, as in Fig. 3b), and $\alpha_w$, $\alpha_g$ and $\rho$ constants (see Methods). This reduced model summarizes that convergence to the rewarded location was dictated both by[1] strong, local attractor dynamics, where the progressive increase in synaptic weights nearby the Hebbian assembly works to destabilize and attract neural activity (Fig. 3f, center, red spot), and[2] weaker, global attractor dynamics due to focalization of the internal goal at the Hebbian assembly (dashed box). Both of these terms required an instantaneous DA-excitability action on a previously DA-plasticity-reinforced Hebbian assembly, as they were negligible in the absence of either DA-plasticity or DA-excitability (Fig. 3f left, right). Overall, under the DA-plasticity+excitability condition, phasic DA signaling induced the transient unfolding of a large and deep BPE basin of attraction (Fig. 3g, left), the subsequent focalization of the internal goal (Fig. 3g, center) and, ultimately, the convergence of the e-mouse to the rewarded location (Fig. 3g, right). Thus, DA-plasticity generated *latent* attractors that allowed only weak local convergence of internal goal and e-mouse positions (Fig. 3h, left). DA-excitability revealed these latent attractors, by amplifying both their depth and width, resulting in strong global convergence (Fig. 3h, center), which was not possible without previous reward learning (Fig. 3h, right).

MAGNet theory thus highlights the effects DA activity can have on ongoing behavior and the necessary conditions for DA to exert such effects. This allowed specific predictions to be tested with reward-seeking behavior in actual mice. Specifically, MAGNet theory predicts that, following an initial reinforcement of a central location, artificially stimulating DA when animals are in the periphery of the environment will increase the cumulative probability of convergence to the rewarded location if DA affects both plasticity and excitability, compared to other conditions (Fig. 4a). This effect would result from the unveiling of a goal-encoding attractor, inducing a sudden "magnetic" effect consisting in i) energization, with increased speed (Fig. 4b, first panel), ii) attraction, with a decrease in the animal's distance to the reward (Fig. 4b, second panel) and iii) a reorientation of their approach angle toward the reward location (Fig. 4b, last panel). Compared to Fig. 1 in which optogenetics DA release caused reinforcement, we expect that randomly stimulating DA in the periphery will avoid cumulating the DA-plasticity effects at the same location (which could create a new attractor) and thus specifically test how DA-excitability reveals previous DA-plasticity effects. Hence, MAGNet theory predicts that manipulating DA will affect animal movements only if there is an attractor in the behavioral energy landscape (Fig. 4c), e.g. in a context in which a central location has been previously rewarded (i.e. when an attractor can be revealed), whereas DA manipulation will not exert any effect on behavior in another (neutral) context (i.e. if there is no attractor to reveal).

## MAGnet predictions are confirmed by dopamine manipulation in mice

We then experimentally tested the predictions (Fig. 4) from the MAGNet model in an equivalent experimental setting (Fig. 5a). In a circular arena, we paired the central location with MFB electrical stimulation (in order to keep the mice naive for future photostimulations, Fig. 5a, Methods) to establish the reward-place association (Fig. 5b), with mice having to leave the location before being stimulated again upon re-entry[23]. As expected, this resulted in a strong enhancement of the central place preference ($F_{(9)} = 5.57$, $p < 1e-16$, Fig. 5b), so the current intensity was adjusted to achieve a moderate visit rate (Supp Fig. 2). This circular arena with an MFB-reinforced central location was considered as the reward (R) context, while a square open-field without any history of reinforcement was considered as the no reward (no-R) context (Fig. 5a vs Fig. 4c). Once the association was learned, we then used VTA photostimulation (Methods, Fig. 5a) to test for MAGNet's predictions on the context-dependent effects of increased DA on movement ballistics (Fig. 4). We provided brief photo-stimulations when mice were away from the central position in the R ("R Chr2 ON") and no-R ("no-R Chr2 ON") contexts. The control conditions (i.e. with baseline DA levels during ongoing decisions) consisted of mice tested in either the reward R and no-R contexts without VTA photostimulations (with 2 experimental groups, YFP ON or Chr2 OFF, to control for light effects or injection effects). VTA photostimulations increasing phasic DA signaling in the environment periphery were provided randomly in space to avoid cumulating the DA-plasticity effects that could eventually create a new rewarded location (Methods).

In ChR2-expressing mice tested in the R context, VTA photo-stimulation decreased the delay to the reward location compared to control times (Fig. 5c,d). This effect was neither observed in YFP-expressing animals, nor in ChR2-transduced animals in the no-R context

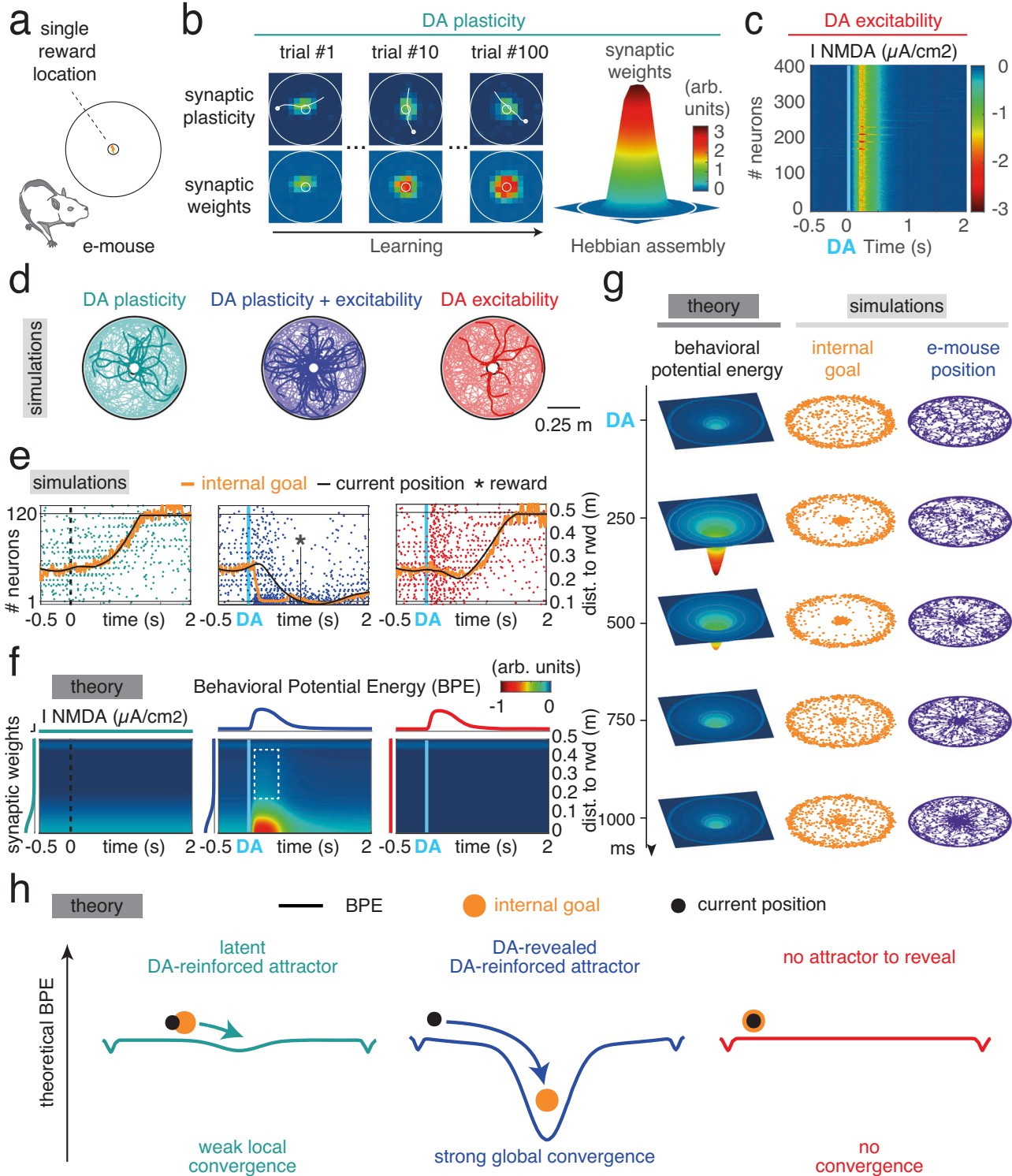

(Fig. 5c,d). Hence, a decrease in the latency to visit the rewarded location was only observed in "R Chr2 ON" animals (Fig. 5c,d), as predicted by MAGNet for the R-context + periphery DA condition (Fig. 4b,c).

We next investigated whether this reduced delay following VTA photostimulation reflected an increase in speed (Fig. 5e,f), i.e. an energizing effect[1,19] rather than simply an increase in the overall pace (frequency) of behavior[38]. VTA photostimulation in the reward context resulted in an increase of animal speed (Fig. 5f), which was not observed in YFP controls (Supplementary Fig. 2). This online effect of VTA photostimulation on speed was consistent with the MAGNet model

(Fig. 4b). Furthermore, VTA photostimulation did not affect speed in ChR2 animals in the no-R context (Fig. 5f). Online manipulation of VTA DA signaling thus affected the speed of action, but only in the context in which a location had been rewarded, consistent with MAGNet prediction (Fig. 4b). Hence, VTA DA signaling only exerted an energization effect in the reward context, which is incompatible with decision-threshold models that predict context-independent speed increases[1,19]. The MAGNet theory, based on attractor dynamics, also predicts that the increase in speed following DA stimulation would be directed towards the reinforced location. We thus assessed whether online VTA DA

**Fig. 3 | Dopamine builds and reveals latent network attractors encoding internal goals. a** Schematics of the single rewarded location arena. **b** Under DA-plasticity alone, phasic DA delivered at the rewarded location yielded long-term synaptic changes (top panels) that accumulated (bottom), eventually shaping a Hebbian assembly encoding the place–reward association (right). **c** Under DA-excitability alone, DA transiently increased synaptic efficacy in the whole network through NMDA potentiation. **d** Superimposed example e-mouse trajectories in the DA-plasticity, DA-excitability and DA-plasticity+excitability conditions, from random positions and directions. Rewarded trajectories are in bold. The color code for conditions is used in panels (e–j). **e** Example model dynamics (neural spiking sorted according to the distance to reward) in the three conditions. Under DA-plasticity +excitability, DA generated a massive neural co-activation at the Hebbian assembly, setting the internal goal at the rewarded location and e-mouse convergence toward

it (reward). The Hebbian assembly was generally unexpressed under DA-plasticity, or absent under DA-excitability, forbidding goal-directed behavior. **f** Theoretical behavioral potential energy (BPE) computed as a function of time and distance to reward under the three conditions. The rewarded location becomes a transient attractor of behavioral dynamics only under DA-plasticity. Faint blue strip at the top reflects the propensity to follow walls during default behavior. **g** Theoretical BPE (illustrated in 2D), as well as internal goal and e-mouse position of example simulations, under DA-plasticity-excitability, during the first second following phasic DA. **h** Schematics of attractorial dynamics in MAGNet. Theoretical BPE were computed at their maximal amplitude after phasic DA in the three conditions. Under DA-plasticity-excitability, navigation toward the reward arises from the convergence toward the BPE minimum, which sets the internal goal.

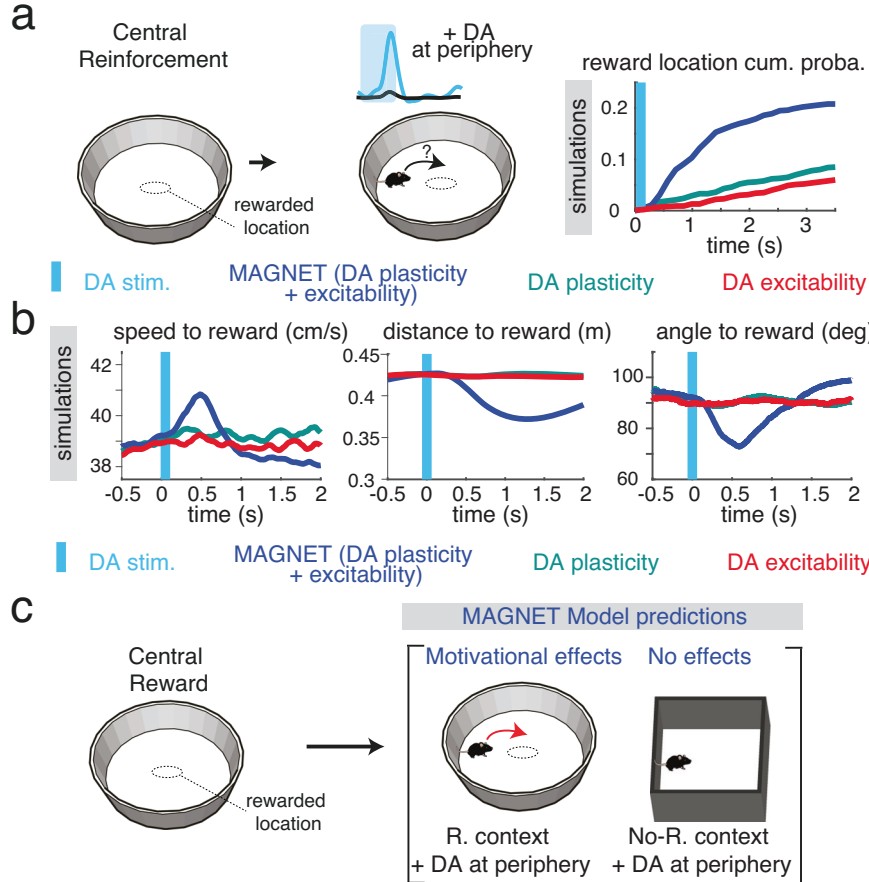

**Fig. 4 | The MAGNET model predicts that DA manipulations exert context-dependent, joint effects on movement direction and ballistics. a** Schematics of conditions used to derive model predictions: following reinforcement, i.e. reward delivery at a central location (left), dopamine is released (vertical blue bar) when the e-mouse is at the periphery of its environment (middle). MAGNet predicts an increased cumulative probability of visiting the reward location only if DA affects both plasticity and excitability, but not if DA solely affects plasticity or excitability

(right). **b** The MAGNet model predicts that the increased pace of visiting the reward location is due to joint effects on movement ballistics, i.e. an increase of speed to reward (left), a decrease of distance to reward (middle) and a reorientation of the angle to the reward (right). **c** The MAGNet model further predicts that the above motivational effects following DA manipulation are specific to the environment in which the central location was rewarded (R. context), and that none of the DA effects on movement ballistics (b) are observed in other contexts (no-R. context).

signaling also affected mice directional behavior. First, the distance between ChR2-transduced animals and the central location (Fig. 5g,h) decreased upon VTA DA photostimulation in the R context but not in the no-R context (Fig. 5h) nor in YFP animals (Supplementary Fig. 2). Second, the accumulated sum of angles between the animal and the goal ("error angle", Fig. 5i) decreased following stimulation in ChR2-expressing animals in the R context (Fig. 5j,k) indicating more direct trajectories to the reward, rather than faster trajectories in any direction. This was neither the case in YFP-expressing mice (Supplementary Figure 2), nor in ChR2 animals in the no-R context (Fig. 5j,k). We further

show that DA effects on behavior (i.e. Figs. 1,5) did not differ between male and female mice (Supplementary Figure 3).

Hence, the increase in animal speed after photostimulation of VTA DA neurons was directed toward the central location, consistent with MAGNet's first predictions (Fig. 4b). VTA DA photostimulation only attracted the animals toward the center location if this location had been previously rewarded, validating the second MAGNet's prediction (Fig. 4c). We further confirmed MAGNet predictions by simulating experimental data from our previous work[13] in which we used photo-inhibition of VTA DA neurons. As predicted by MAGNet

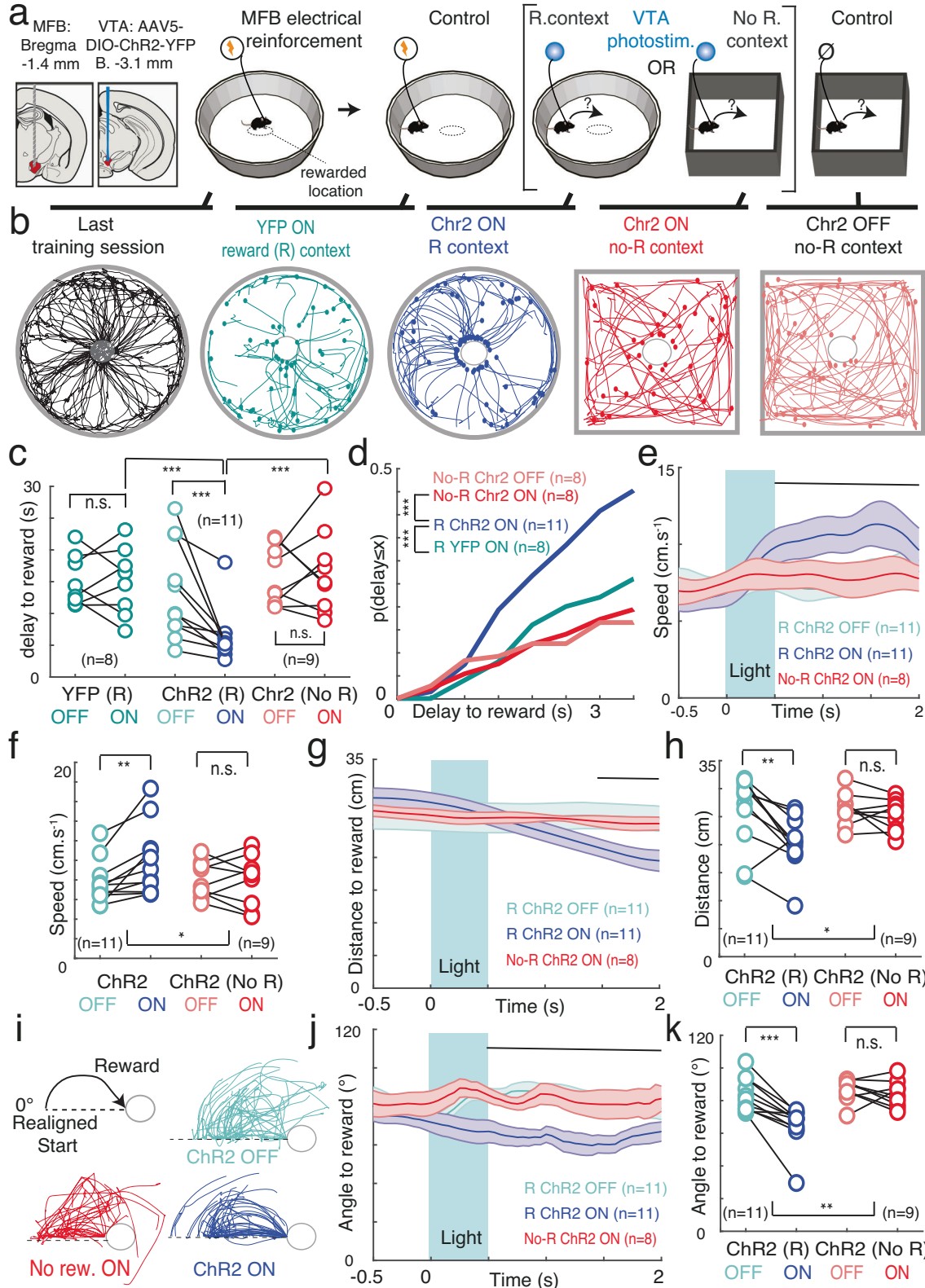

(Supplementary. Figure 4), VTA DA photo-inhibition slowed down the speed of mice in a reward context (i.e. an open-field with three locations previously rewarded with MFB electrical stimulation), but did not affect speed in a non-reward context (homecage[13]).

Overall, our results suggest that instantaneous DA-excitability (motivation) acts in a content-specific and context-dependent manner to retrieve the goal learned under DA-plasticity (reinforcement).

## MAGNet theory reconciles previous conflicting results on motivational dopamine

Finally, we used a last prediction from MAGNet to reconcile the seemingly contradictory literature on the effects of optogenetic DA manipulation on ongoing behavior. MAGNet predicts that goal-directed convergence increases from relatively distal positions upon DA release at the periphery (when the initial position is far from the

**Fig. 5 | Testing the prediction that VTA photostimulation-induced movements are goal-specific and context-dependent. a** Schematics of electrode implantation in the medial forebrain bundle (MFB) and injection of the ChR2-YFP-expressing virus and fiber implantation in ventral tegmental area (VTA). **b** Experimental test of the model predictions. A location is rewarded by MFB electrical stimulation (left). Then (inside the brackets), VTA photostimulation is provided in the context where reinforcement occurred (R. context) and in another context (no R. context, where no location had been rewarded). Each condition is compared to controls without acute VTA photostimulation (for R. context : MFB+YFP animals ON light, and MFB+Chr2 animals OFF light; for No R. context : Chr2 animals OFF light). **c** From left to right : example trajectories at the end of the MFB conditioning sessions, and post-photostimulation bouts of trajectories in the different conditions described in **c**. Differences between photostimulation-rewarded location delays for YFP (reward context, R/YFP ON versus OFF, paired t-test $T_{(7)}= -0.07$, $p = 0.94$); ChR2-expressing (reward context, $T_{(10)}= -3.58$, $p = 0.05$) and ChR2-expressing (no reward context, $T_{(8)}= 0.32$, $p = 0.76$) animals. **d** Cumulative distribution of the photostimulation-rewarded location delays in YFP (ON versus OFF light in reward context, Kolmogrov–Smirnov test on all trials from all mice: $p = 0.23$); ChR2-expressing

(Reward context, KS test: $p = 1.10^{-8}$) and ChR2-expressing (no-Reward context, KS test: $p = 0.81$) animals. **e–k**. Speed (**e**), distance to the rewarded location (**g**), and angle between the animal and the rewarded location **j** around VTA photostimulation (blue shading) for ChR2-expressing animals when ON light in reward context (purple), OFF light in reward context (light blue) and ON light in no reward context (red). Average difference in speed (**f**), distance to the rewarded location (**h**), and angle between the animal and the rewarded location (**k**) between ON and OFF light conditions, in reward ("R-ChR2", ON versus OFF paired t-test after stimulation for speed: $T_{(10)}=3.46$, $p = 0.006$, distance: $T_{(10)}=-3.68$, $p = 0.004$, angle: $T_{(10)}=-5.32$, $p = 3.10^{-4}$) and no reward ("No-R ChR2", speed : $T_{(8)}=-0.17$, $p = 0.87$, distance: $T_{(8)}=-1.17$, $p = 0.27$, angle: $T_{(8)}=-0.89$, $p = 0.40$) contexts. **i** shows the computation of angle between the animal and the rewarded location, based on the same trajectories as in (**a**), realigned to the same line relative to the rewarded location, showing straight trajectories for animals when ON light in reward context (purple), and indirect trajectories when OFF light in reward context (light blue) or ON light in no reward context (red).Data are presented as mean ± s.e.m. See also source data file.

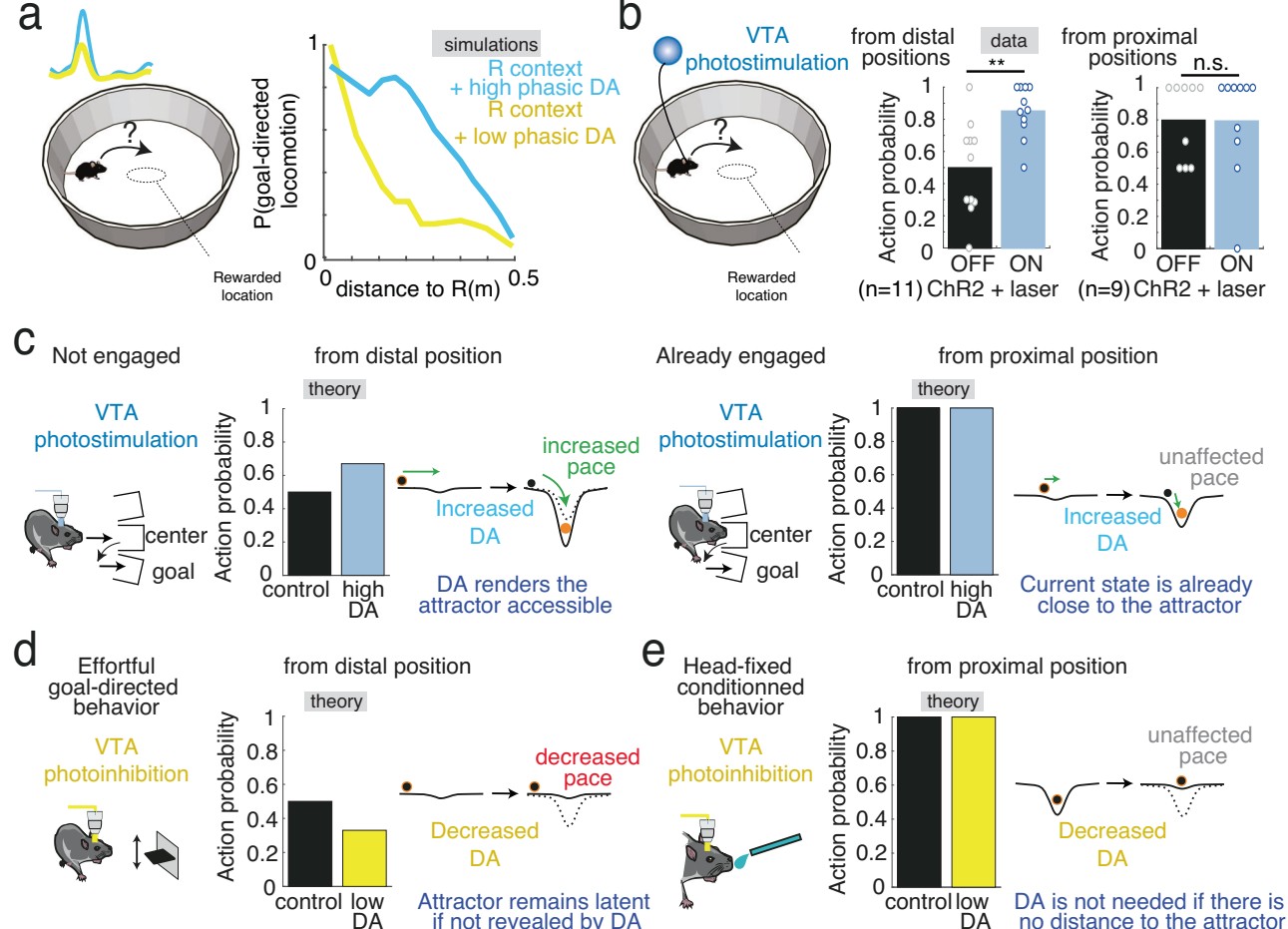

**Fig. 6 | MAGNet provides a framework reconciling conflicting results on the motivational role for dopamine. a** In MAGNet simulations, peripheral DA release after the reinforcement of a central location affects the probability that the e-mouse directs towards the central location (i.e. the goal-encoding attractor), compared to a baseline DA level. MAGNet simulations predict that such an increase in goal-directed locomotion depends on the distance between the initial state and the attractor state. **b** Experimental data show that VTA DA photostimulation in the Reward context affects the probability that mice direct toward the central location when mice are initially away from the central location (paired t-test ON versus OFF : $T_{(10)}=4.20$, $p = 0.0018$) but not when VTA DA photostimulation occurs in proximal locations ($T_{(8)}=1.13$, $p = 0.29$). **c** Simulation of Hamid et al. data (port-choice task) with the MAGNet reduced model. The action latency after DA stimulation decreases

only for animals not already engaged in the task. This differential increase in action pace is accounted for by the reduced model where increased action probability at high DA only occurs when considering a distant (versus proximal) initial state (here, animal position) from the goal-encoding attractor situated at the rewarded location. **d** Simulation of Fischbach-Weisset et al. data (leverpress task) with the MAGNet reduced model, in which action probability decreases after VTA DA inhibition when the initial state is distant from the goal-encoding attractor. **e** Simulation of Lee et al. and Coddington & Dudman (head-fixed licking task) with the MAGNet reduced model, in which the action probability does not decrease after DA inhibition when the initial state is near or upon the goal-encoding attractor. Data are presented as mean ± s.e.m. See also source data file.

attractor), but not from positions closer to the central location (when the initial position is already close to the attractor, Fig. 6a). In our experimental data, we confirmed that the probability to go towards the central location was not affected when the VTA DA photo-stimulation occurred on proximal positions, but was increased for distal VTA DA photo-stimulation (Fig. 6b). In this context of investigating the "distance-to-attractor"-dependent effects of DA on ongoing behavior, we then assessed how MAGNet re-interprets previous experiments. Among the mixed evidence for a motivational role for DA, Hamid et al.[8] found that optogenetic phasic activation of DA neurons shortened the latency for rats to engage in a reward-related task, but only when the rat was not already engaged in the goal-directed behavior. The reduced MAGNet model accounts for this increased action pace, as the effects of increased phasic DA signaling on the following movements depend on the animal's distance to the goal (Fig. 6c), i.e., the action probability is only increased when the initial location is distant from the goal-encoding attractor. The effects of optogenetic inhibition of DA neurons on ongoing behaviors has also shown conflicting results. Fischbach–Weisset et al.[39] and Bousseyrol et al.[13] found that decreased phasic DA signaling delayed goal-directed actions (lever press and locomotion, respectively), while Lee et al.[16] and Coddington & Dudman[10] found the opposite, i.e., that decreased phasic DA signaling did not exert any effect on head-fixed behavior. Like for Hamid et al.[8] data, the MAGNet model allows interpreting this discrepancy in terms of distance between the initial state and the goal-encoding attractor. Actions associated with a high initial distance to the attractor are sensitive to decreased phasic DA signaling (Fig. 6d). However, when the animal is already in close proximity to its goal (as we assume to be the case for head-fixed animals, which do not need to move or exert efforts to get a reward), decreasing phasic DA signaling has no effect. In head-fixed behavior, it can be considered that the mouse initial location lies in the vicinity of the goal-encoding state (Fig. 6e). MAGNet further predicts that the magnitude of DA manipulation and the distance to the goal interact in a non-linear manner to affect action probability, latency and speed (Supplementary Fig. 5). Hence, MAGNet provides a dynamic biophysical ground and theoretical framework for the concept that high DA is not needed when an action is underway or when the goal is nearby, but crucial for a flexible approach toward distant, non-trivial goals[25].

## Discussion

MAGNet theory interprets goal-directed actions as a two-step process: neural assemblies representing a potential goal are learned through synaptic plasticity regulated by reward-signaling dopamine (DA), but not systematically expressed, i.e. they constitute latent[40] attractors in a behavioral energy landscape. Then, spontaneous or reward-predicting phasic DA neuromodulation renders these attractors accessible from distal starting positions, by widening and deepening their basins of attraction. We validated some of the predictions from the MAGNet theory experimentally, using optogenetics, showing that online phasic VTA DA signaling immediately orients the animal toward rewarded locations and energizes specific, context-dependent actions previously associated with phasic VTA DA. MAGNet also reconciles conflicting experimental evidence regarding whether phasic VTA DA affects ongoing behaviors or not[8,10,13–16,39], based on the distance between the animal state and its potential goal's basin of attraction.

### How the MAGNet theory differs from other theories of dopamine function

Originally, reinforcement learning theory did not assign any effect to DA during ongoing behavior, once the value of actions has been learned[2,3]. DA has then been suggested to exert either (a) directional effects towards a specific goal, by itself[17,21] or through stimulus-driven DA release that directs the behavior toward a cue[31,41] or (b) activational effects, i.e. with DA increasing the probability and vigor of any motor

behavior[1,19,20]. Our theory proposes that DA exerts joint directional and activational effects, but only in contexts where a goal-encoding attractor exists in the behavioral landscape.

DA nuclei do not seem to have enough encoding capacity, and DA projections are not selective enough[1,22], for a precise directional role, even though it is correlated with broad (e.g. forward or backward) movement directions[21]. Alternatively, DA is proposed to add incentive salience to the stimulus cue being currently processed, promoting approach. The DA-associated cue is described in incentive-salience accounts as becoming "magnetic"[41], which is exactly what is expected in MAGNet theory for a state suddenly attracting the decision network's dynamics. However, actions that are not cue-driven but self-generated rely on internal representations, in which case the role of DA in incentive-salience is less specified. Our proposal generalizes the concept of incentive motivation by suggesting that it involves making goal-encoding attractors (either cued or internally generated) accessible. In this framework, DA does not play a directional role. Instead, the chosen goal depends on the reward context and is not specified by DA release. This contrasts with Pavlovian conditioning, where the stimulus cue triggers both phasic DA activity and approach. In our theory the stimulus cue sets the context (the latent attractor) and triggers a phasic DA activity prompting convergence in that energy landscape. This would explain why cue-triggered DA is needed for animals to display sign-tracking[31], but optogenetic activation of DA neurons outside the context cannot replace the stimulus cue[15], because there is no attractor to reveal (similar to the No-Reward condition in the present study).

Activational accounts of DA assign a role to phasic DA in gating decisions and/or energizing actions. In "incentive motivation" modified RL models, phasic DA could increase the probability to accept decision[23], and in drift-diffusion models, phasic DA has been suggested to move the decision threshold[1,19]. These models predict an increase in the probability of all actions following VTA photostimulation, in opposition to our data showing an absence of DA effects outside the Reward context. Furthermore, we show that angle and speed profiles, not just latency or average speed, are affected by phasic DA, which go beyond the scope of these models (and of other theories considering action latency or frequency[18,38,42]). SNc stimulation has been found to induce context-independent effects on locomotion[11] contrary to our VTA data. This discrepancy could arise from VTA building and expressing high-level goals (substantially separated wells in the energy landscape), while SNc would build and express low-level, context-independent subgoals (i.e. motoric action) corresponding to multiple nearby attractors.

In the context of working memory, tonic levels of prefrontal DA may maintain persistent activity encoding a goal[37,43]. In this account, D2R favors stimulus-driven transitions toward another state by rendering attractors more shallow, while the current state is stabilized by D1R-mediated deepening of its basin of attraction. This model differs from ours, in which phasic DA activates D1R to widen basins of attraction, setting a new goal. DA may achieve a "double duty" in cognitive motivation[24] by widening (to promote a decision) and deepening (to stabilize its working memory) basins of attraction. Furthermore, the present experimental test considered the physical space, but the conceptual consequences of MAGNet extend to non-physical spaces with more abstract goals and task structures[24,25].

### Biophysical implementation of the MAGNet theory

Our biophysical implementation of MAGNet, although not explicitly tested here, is derived from widespread findings from the literature. The attractorial principle of MAGNet is consistent with brainwide attractor dynamics; e.g. in the frontal cortex[44], but also in premotor[45], visual[46] or limbic[47] structures. The current implementation of MAGNet relies on a recurrent network with cortical connectivity, but other implementations are possible, e.g. with cortico-striatal loops, given the

known importance of mesolimbic DA for approaching rewards[20,25]. In basal ganglia models, navigation toward goals can be learned through reinforcement-learning of synapses between space- and action-coding (striatal) neurons[48]. Other basal ganglia models have proposed a link between action selection and action intensity[49], accounting for some of the roles of basal ganglia in energizing behaviors[19]. DA regulation on both synaptic plasticity and excitability could result in multiplicative effects of DA[50] on action selection and energization in a (yet to be achieved) striatal model combining these features. However, navigation in models of basal ganglia corresponds to the animal progressively following gradients of space-action values[48], analogous to the local convergence along synaptic weight gradients in the MAGNet model. Hence, it seems more difficult for current basal ganglia models to account for the widening of the goal's basin of attraction, which requires a distant signal focalizing the dynamics of the internal goal, from any initial condition. Finally, DA effects on the amygdala, thalamus and hippocampus[19,34], would require a full-scale modeling (being out of scope). We thus lumped some of the decision processes into simple (e.g. spatial coding as a topographical input) or algorithmic descriptions (e.g., motor convergence as ballistics commands).

At the cellular level, we focused on NMDA modulation by DA at both the plasticity (long term) and excitability (short term) levels, but DA can also affect a vast diversity of receptors and ionic channels, depending on DA receptors[36,37]. Here we mainly modeled D1R effects to account for approach behaviors, but D2R may not be as antagonistic to movement as previously believed: D1R and D2R may actually be synergistic for cortical plasticity, when considering the cAMP-PKA pathway we considered[36]. For the regulation of intrinsic excitability, D2R may exert destabilizing influences (rather than inhibitory) that promote or oppose D1R effects depending on down or up-states, respectively[36]. These interactions hint at complementary roles in our dynamical framework. Another important cellular feature of MAGNet concerns plasticity pathways implementing eligibility traces. We followed the recent literature describing two distinct eligibility traces for LTP and LTD[4]. Early LTP and LTD are believed to depend on CaMKII and calcineurin, respectively, while in the present model a different couple of kinase and a phosphatase is needed for LTP and LTD. This may be implemented by compartmentalization via synaptic scaffolds linking different forms of CaMKII with different phosphatases[51]. Downstream decoding of early LTP/D may be achieved by ERK and CREB[52,53]. In MAGNet, DA is key to transform eligibility traces into effective plasticity, but other neuromodulators such as noradrenaline, serotonin and acetylcholine seem to exert differential effects on the read-out of LTP and LTD[4].

Nevertheless, the reduced model we provide (Behavioral Potential Energy) does not depend on the free parameters from the biophysical network model. Future work could aim at reexpressing the MAGNet theory in RL formalism, but this would require accounting for the non-linear synergy between DA-plasticity and DA-excitability effects we unravel.

## "Latent attractor" as a dynamical framework distinguishing learning from performance

Self-generated actions have proven hard to account for in classical attractor models. In such models, neural state transitions from spontaneous activity to decision attractors[27] may be triggered by a destabilizing stimulus[28]. Alternatively, spontaneous state and decision state may coexist as distinct, stable attractors[21,32], with neural fluctuations driving transitions toward decisions. However, goal-directed decisions are neither random nor necessarily cue-triggered[29,40]. Rather, they are self-generated, based on internal action-outcome representations[29]. More refined models consider partially stable attractors, allowing dynamics to eventually escape and converge to another attractor[54–57]. This requires specific mechanisms, either synaptic inhibition designed to repel the neural dynamics from the attractor[56] or neuronal fatigue ensuring the attractor to be only transient once activated[57]. Contrary to these models, the decision attractor simply vanishes in MAGNet, once the excitability effect of phasic DA decays due to DA recapture. Hence, in our theory, both entering into, as well as exiting from, a decision attractor are controlled by an internal operation.

Such internal control also effectively decouples the neural dynamics from synaptic changes, which is key to account for goal-directed actions. Usually, reward-dependent synaptic plasticity directly leads to a change in models' neural dynamics, yielding behavioral adaptation (i.e. change in the frequency of behavior). However, animals do not always express learning as behavioral changes. Instead, some forms of learning are latent[29,40]. For instance, a sated animal may learn to navigate a labyrinth containing a food source without increasing the visits to the food source, and, upon food deprivation, display a change in its behavior (i.e. going to the food source). MAGNet accounts for such latent learning by DA-modulated synaptic plasticity building latent attractors that do not necessarily affect neural dynamics. MAGNet theory decouples learning and performance because it considers the dynamical convergence in the joint (i.e. cartesian product of) neural and behavioral spaces. DA exerts a distant, discontinuous role that widens the decision's basin of attraction, so that the internal goal can be instantaneously set at a goal distant from the initial position.

Nevertheless, in the MAGNet theory, phasic DA should not be mistaken for a "homunculus" taking the decision to move. While we focused on the effect of DA rather than on the origin of phasic DA (which we considered here triggered either by a reward, spontaneous, or manipulated externally), the question of how phasic DA occurs in self-paced actions remains open. Our theory can combine with time-difference accounts reward-prediction errors[2,3], in which the reward prediction term (that would be observed at the beginning of self-paced movements) would be used for initiating and controlling goal-directed actions.

Overall, this study is in line with the current paradigmatic shift regarding neurodynamics[58]: instead of being permanently attracted by Hebbian attractors[27,28], collective dynamics within neural circuits may rather be governed through latent attractors controlled by context-related phasic neuromodulation, thus expressing specific, learned goal-directed actions only in certain brain states.

# Methods

## Animals

Experiments were performed on DAT[iCRE] female ($n = 26$) and male ($n = 21$) mice, from 8 to 16 weeks old, weighing 25–35 g. Mice were housed in cages in an animal facility on a 12 h light/dark cycle where the temperature ($20 \pm 2$ °C) and humidity were automatically monitored, with food and water available ad libitum. DAT[iCRE] mice[59] were kindly provided by Ludovic Tricoire and genotyped by PCR as described previously[60]. No statistical methods were used to predetermine sample sizes, which are comparable to previous studies[13,32,33] using similar techniques and animal models.

## Ethics statement

All experiments were performed in accordance with the recommendations for animal experiments issued by the European Commission directives 219/1990, 220/1990 and 2010/63, approved by Sorbonne University, and n° 014378.01 supervised by the CEEA – 005.

## Virus production

AAV vectors were produced as previously described[61] using the co-transfection method, and purified by iodixanol gradient ultracentrifugation[62]. AAV vector stocks were titrated by quantitative PCR (qPCR[63],) using SYBR Green (Thermo Fischer Scientific).

## Virus injections

Mice were anesthetized with a gas mixture of oxygen (1 L/min) and 1–3 % of isoflurane (Piramal Healthcare, UK), then placed into a stereotaxic frame (Kopf Instruments, CA, USA). After the administration of an analgesic (Buprecare 0,1 mL at 0,015 mg/L) and of a local anesthetic (Lurocain, 0.1 mL at 0.67 mg/kg), a median incision revealed the skull which was drilled at the level of the VTA. Mice were then injected unilaterally in the VTA (1 µL, coordinates from bregma: AP −3.1 mm; ML ± 0.5 mm; DV −4.5 mm from the skull) with an adeno-associated virus (AAV5.Ef1a.DIO.ChR2.YFP 6.89e13 vg/mL or AAV5.Ef1a.DIO.YFP 9.10e13 vg/mL)[33]. A double-floxed inverse open reading frame (DIO) allowed to restrain the expression of ChR2 to VTA dopaminergic neurons. After stitching and administration of a dermal antiseptic, mice were then placed back in their home-cage and had 14 days to recover from surgery.

## Fiber and electrode implantations

Two weeks after virus injection, mice were anesthetized as above. After the administration of the analgesic and local anesthetic, skin was incised, the skull was drilled at the level of the VTA. An optical fiber (200 µm core, NA = 0.39, Thor Labs) coupled to a ferule (1.25 mm) was implanted just above the VTA ipsilateral to the viral injection (coordinates from bregma: AP −3.1 mm, ML ±0.5 mm, DV 4.4 mm), and fixed to the skull with dental cement (SuperBond, Sun Medical).

For dual implantation experiments, the skull was also drilled at the level of the Median Forebrain Bundle (MFB). A bipolar stimulating electrode was then implanted unilaterally (ipsilateral to the optic fiber in the VTA) in the brain (stereotaxic coordinates from bregma according to Paxinos atlas: AP −1.4 mm, ML ±1.2 mm, DV −4.8 mm from the brain)[33].

After stitching and administration of a dermal antiseptic, mice were then placed back in their home-cage and had 14 days to recover from surgery. The behavioral task began 4 weeks after virus injection to allow the transgene to be expressed in the target dopamine cells.

## Ex vivo patch-clamp recordings of VTA DA neurons

To verify the functional expression of the excitatory opsin ChR2, 8–12 week-old male and female DATiCRE mice were injected with the ChR2-expressing virus as described above. 4 weeks after infection, mice were deeply anesthetized with an intraperitoneal (IP) injection of a mix of ketamine/xylazine. Coronal midbrain sections (250 µm) were sliced using a Compresstome (VF-200; Precisionary Instruments) after intracardial perfusion of cold (4 °C) sucrose-based artificial cerebrospinal fluid (SB-aCSF) containing (in mM): 125 NaCl, 2.5 KCl, 1.25 NaH$_2$PO$_4$, 5.9 MgCl$_2$, 26 NaHCO$_3$, 25 Sucrose, 2.5 Glucose, 1 Kynurenate (pH 7.2, 325 mOsm). After 10–60 min at 35 °C for recovery, slices were transferred into oxygenated aCSF containing (in mM): 125 NaCl, 2.5 KCl, 1.25 NaH$_2$PO$_4$, 2 CaCl$_2$, 1 MgCl$_2$, 26 NaHCO$_3$, 15 Sucrose, 10 Glucose (pH 7.2, 325 mOsm) at room temperature for the rest of the day and individually transferred to a recording chamber continuously perfused at 2 ml/min with oxygenated aCSF. Patch pipettes (4–8 MΩ) were pulled from thin wall borosilicate glass (G150TF-3, Warner Instruments) using a micropipette puller (P-87, Sutter Instruments, Novato, CA) and filled with a potassium gluconate (KGlu)-based intra-pipette solution containing (in mM): 116 K-gluconate, 10–20 HEPES, 0.5 EGTA, 6 KCl, 2 NaCl, 4 ATP, 0.3 GTP and 2 mg/mL biocytin (pH adjusted to 7.2). Transfected VTA DA cells were visualized using an upright microscope coupled with a Dodt contrast lens and illuminated with a white light source (Scientifica). A 460 nm LED (Cooled) was used both for visualizing YFP-positive cells (using a bandpass filter cube) and for optical stimulation through the microscope (with same parameters used for behavioral experiments: ten 5-ms pulses at 20 Hz). Whole-cell recordings were performed using a patch-clamp amplifier (Axoclamp 200B, Molecular Devices) connected to a Digidata (1550 LowNoise acquisition system, Molecular Devices). Signals were low-pass filtered

(Bessel, 2 kHz) and collected at 10 kHz using the data acquisition software pClamp 10.5 (Molecular Devices). All the electrophysiological recordings were extracted using Clampfit (Molecular Devices) and analyzed with R.

## Behavior acquisition and conditioning procedures

Experiments were performed using a video camera connected to a video-track system, out of sight of the experimenter. No exclusion criterion was used except for prior MFB electrical stimulation (see below). Behavioral conditioning and test sessions were independently replicated by two different experimenters (JN, BH). Experimenters were blind to the condition (YFP or Chr2) at the time of the behavioral tests. A home-made software (Labview National instrument) tracked the animal, recorded its trajectory (20 frames per s) for 10 min and sent TTL pulses to the electrical stimulator or LED device when appropriate[32,33].

Conditioning procedure with VTA DA photostimulation: three explicit square locations, marked on the floor, were placed in a circular open-field (67 cm diameter), forming an equilateral triangle (side = 35 cm). Each time a mouse was detected (by its centroid) in the area of one of the rewarding locations (area radius = 3 cm), a 500-ms train of ten 5-ms pulses at 20 Hz was delivered to the LED device. An ultra-high-power LED (470 nm, Prizmatix) coupled to a patch cord (500 µm core, NA = 0.5, Prizmatix) plugged onto the ferule was used for optical stimulation (output intensity of 10 mW). Animals could not receive two consecutive stimulations in the same location.

Conditioning procedure with MFB electrical stimulation: only one explicit location was marked on the floor, at the center of the open-field. Each time a mouse centroid was detected in the area (radius = 5 cm) of the location, a 200-ms train of twenty 0.5-ms biphasic square waves pulsed at 100 Hz was delivered to the electrical stimulator. Mice were required to leave the location (i.e. to be detected at least 10 cm from the central point) for the stimulation to be made available again. The training consisted of a block of 5 daily sessions of 10 min at 80 µA (2 mice not self-stimulating at least 50 times in 10 min were excluded at this stage), followed by 5 daily sessions of 10 min in which ICSS intensity was adjusted (in a range of 20–200 µA) so that mice visited the central location between 20 and 50 times at the end of the training.

Test sessions with VTA DA photostimulation: after the end of the MFB electrical conditioning procedure, the optical stimulation patch cord was plugged onto the ferule during at least one OFF day (maximum = 5) to habituate the animals, until the criterion (between 20 and 50 locations visits in 10 min) was reached again. On ON test days, photostimulation was delivered by the experimenter when the animal was outside of the reinforced location (at least 10 cm from the central point). When the experimenter clicked to stimulate, it had a 50% probability to deliver an actual TTL pulse leading to photostimulation, otherwise this time point was recorded as a control. In the square open-field test, occurring after the test session in the circular open-field, the procedure was the same, except that it took place in square open-field (side = 70 cm) without any mark on the center.

## Behavioral analyzes and statistics

Stimulation-reward duration was computed as the time between the start of the photostimulation (or of the control time) and the first detection of the animal in the central location. Durations greater than 60 s were excluded from the analysis for the sake of representations, but did not affect the statistical significance of the tests. Cumulative distributions of durations were computed by pooling stimulation-reward and control time-reward from all animals in one condition (e.g. ChR2 or YFP), with a 3-s time bin. Average delays to rewards were also computed for each animal. For all groups of mice, the trajectory was smoothed using a triangular filter before computing the instantaneous speed, which corresponds to the distance traveled by the animal between two video frames (every 50 ms) as a function of

time. Mean acceleration following stimulation was taken as the time derivative of speed during the first second after stimulation. Angles to reward were computed as the angles between each successive position of the animal relative to the initial angle (at photostimulation or at control time). Angle error was taken as the mean of $||\Sigma e^{i\theta}||$ where $\theta$ are the successive angles to reward.

All statistical analyzes were computed using Matlab (2023) with custom programs. Results were plotted as a mean ± s.e.m. The total number ($n$) of observations in each group and the statistics used are indicated in figure legends. Classical comparisons between means were performed using parametric tests (Student's T-test, or ANOVA for comparing more than two groups) when parameters followed a normal distribution (Shapiro test $P > 0.05$), and non-parametric tests (here, Wilcoxon or Mann-Whitney) when the distribution was skewed. All tests were two-sided. Repeated-measure ANOVAs were used for longitudinal measures. Multiple comparisons were Bonferroni-corrected.

### Immunochemistry

After euthanasia, brains were rapidly removed and fixed in 4% paraformaldehyde (PFA). After a period of at least three days of fixation at 4 °C, serial 60-μm sections were cut with a vibratome (Leica). Immunostaining experiments were performed as follows: VTA brain sections were incubated for 1 hour at 4 °C in a blocking solution of phosphate-buffered saline (PBS) containing 3% bovine serum albumin (BSA, Sigma; A4503) (vol/vol) and 0.2% Triton X-100 (vol/vol), and then incubated overnight at 4 °C with a mouse anti-tyrosine hydroxylase antibody (anti-TH, Sigma, T1299) at 1:500 dilution, in PBS containing 1.5% BSA and 0.2% Triton X-100. The following day, sections were rinsed with PBS, and then incubated for 3 h at 22–25 °C with Cy3-conjugated anti-mouse and secondary antibodies (Jackson ImmunoResearch, 715-165-150) at 1:500 in a solution of 1.5% BSA in PBS, respectively. After three rinses in PBS, slices were wet-mounted using Prolong Gold Antifade Reagent (Invitrogen, P36930). Microscopy was carried out with a fluorescent microscope, and images captured using a camera and analyzed with ImageJ.

Identification of the transfected neurons on DAT$^{iCRE}$ mice by immunohistofluorescence was performed as described above, with the addition of 1:500 Chicken-anti-GFP primary IgG (ab13970, Abcam) in the solution. A Goat-anti-chicken AlexaFluor 488 (1:500, Life Technologies) was then used as secondary IgG. Neurons labeled for TH in the VTA allowed to confirm their neurochemical phenotype, and those labeled for GFP to confirm the transfection success.

### Overview of the biophysical model and phenomenological model

The details of the models can be found in the Supplementary Methods and the code (in Matlab 2018) available in the repository https://zenodo.org/records/13814481. In short, at its largest scale, the e-mouse biophysical model was designed as a distributed decision architecture deciding how an e-mouse navigates in a space. To fit experimental paradigms, we considered the physical space (a circular arena), but the model could extend to any task space. The e-mouse navigation was governed by linear speed and angular commands ensuring convergence toward either a default objective (circling along arena walls) or goal-directed behavior toward an internal goal, set by a recurrent prefrontal neural circuit. The contribution of default behavior to speed was high when the e-mouse headed toward, or was aligned with, the arena walls, but vanished when the e-mouse was far from, or not aligned with, the arena walls. Angular dynamics toward the default objective ensured that the e-mouse aligned with the wall when approaching it. Far from walls, angular dynamics were essentially influenced by goals situated in its visual foreground landscape. The internal goal was determined according to a probabilistic soft-max process (modeling basal ganglia operations), which stochastically selected the preferred position of neurons according to probabilities based on their instantaneous spiking rate. Neuronal preferred positions were organized on a square lattice that covered the arena.

The local recurrent prefrontal network consisted in a detailed biophysical model of PFC neurons and connections[64]. The model contained neurons that were either excitatory (E) or inhibitory (I), with sparse connectivity, an E/I ratio of 4, and E/I current balance at the post-synaptic neuron level. Leaky integrate-and-fire (LIF) neurons were endowed with recurrent and feed-forward currents, and with adaptive action potential threshold in excitatory neurons. Feed-forward currents consisted of AMPA currents while recurrent currents consisted of AMPA, NMDA, GABA-A and GABA-B currents. We considered a uniform delay for synaptic conduction and transmission. AMPA feed-forward currents consisted in two parts: 1) inputs from external sources (putatively sub-cortical and/or cortical inputs), modeled as an exponentially-filtered normal stochastic process with temporally homogeneous mean], and 2) hippocampal place-field inputs encoding the e-mouse position, with PFC neurons receiving input currents proportional to the activation of their receptive fields by a Gaussian input centered on e-mouse position. Recurrent NMDA currents were subject to dopamine modulation that affected their maximal conductance, in all synapses of the network ("DA-excitability").

Network excitatory synapses underwent a dopamine (DA)-modulated form of Hebbian Spike Timing-Dependent Plasticity (STDP) ("DA-plasticity"), with pre- then post-synaptic spike sequences leading to long-term potentiation (LTP), and post- then pre-synaptic spike sequences to long-term depression (LTD). Spiking activity patterns did not translate into immediate effective synaptic changes, but rather resulted in synaptic tags, called eligibility traces[4,5], which were read out at the time of dopamine release[6]. Eligibility traces (eLTP and eLTD, respectively) arose from synaptic calcium dynamics in the post-synaptic button[64,65]. Synaptic calcium took into account the sum of calcium contributions arising from pre- and post-synaptic spiking, together with buffering and extrusion. Intracellular calcium activated calcium-dependent kinases and phosphatases, which competed to form eLTP and eLTD traces. Dopamine gated the transformation of eLTP and eLTD traces into actual changes in excitatory synaptic weights. Dopamine level was the same at all synapses. Dopamine was released when the e-mouse was detected inside rewarded areas, but also occurred spontaneously[9] according to a Poisson process, i.e. with homogenous release probability within each time bin. The dopamine concentration followed second-order dynamics modeling release and recapture.

### Reporting summary

Further information on research design is available in the Nature Portfolio Reporting Summary linked to this article.

## Data availability

A table summary of the experimental data is in the supplementary material (Source Data File). Full behavioral data (tracking of position, timestamps for photostimulation) datasets generated during the current study are available in the repository https://doi.org/10.5281/zenodo.13842201. Source data are provided with this paper.

## Code availability

The Matlab codes for simulating the model and theory can be accessed at https://zenodo.org/records/13814481.

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

## Acknowledgements

We thank Deniz Dalkara and the viral core facility at the Vision Institute (Paris) for the AAV vectors. We thank Lauren Reynolds and Benoît Girard for careful reading of the manuscript. This work was supported by Centre national de la recherche scientifique (CNRS UMR 8249), Fondation pour la recherche médicale (FRM DEQ2013326488 to PF, Fourth year PhD fellowship FDT201904008060 to SM), French National Cancer Institute Grant TABAC-16-022, SPAV1-23, French state funds managed by Agence Nationale de la Recherche (ANR-16 Nicostress to PF, ANR-20 NICADO, ANR-23 VarSeek to PF and BD, ANR-22 LEARN to JN), and Labex memolife (to PF and EV).

## Author contributions

Conceptualization: JN BD PF Methodology: JN BD PF AM Investigation: JN MS SM BH FA EV BD Visualization: JN MS BD Funding acquisition: PF Writing – original draft: JN BD MS PF Writing – review & editing: JN BD MS PF AM SM.

## Competing interests

The authors declare no competing interests.
