## [Peer Review File · Nature Communications]

REVIEWER COMMENTS

Reviewer #1 (Remarks to the Author):

I am happy with the revision, and I recommend publication.

Reviewer #2 (Remarks to the Author):

As an additional reviewer, my task is to evaluate the manuscript by Naudé et al., considering the feedback from other reviewers. While I acknowledge the comprehensiveness of the model presented, I concur with previous reviewers that it seems experimentally underdeveloped. The authors have mainly addressed prior comments by removing the original RL models and incorporating simulations derived from other studies, which I find reasonable but insufficient for fully addressing the raised concerns. Furthermore, I found the main text challenging to comprehend, requiring multiple readings to fully grasp the content. Thus, the readability could be significantly improved.

Overall, I am neutral about this manuscript at this stage. Although unifying the role of dopamine within a theoretical framework is appealing, the current version lacks important experiments and analyses necessary to better align the experimental data with the model. The manuscript could become suitable for publication once these issues are addressed.

1. The experimental validations of the model's predictions seem insufficient. For instance, the manuscript does not elucidate how dopamine is released in their environment. After the initial learning phase (reinforcement), do dopamine neurons become active when mice approach the rewarded location more quickly? Does inhibition of dopamine neurons slow down the speed of mice after the reinforcement? Addressing some of these is important in strengthening the manuscript.
2. Figure 5 addresses the conflicting results regarding the motivational role of dopamine. The authors mention in page 15 line 6, that "... as the effects of increased phasic DA signaling on the following movements depend on the animal distance to the goal ...". It would be more convincing to directly test this hypothesis using their behavior data with VTA photostimulation.

3. The distinction between VTA photostimulation in Figure 1 and Figure 4 is unclear. In Figure 1, photostimulation appears to affect both synaptic plasticity (reinforcement) and excitability (motivation), whereas in Figure 4, photostimulation seems to specifically control excitability (motivation). What is the physiological basis for these assumptions?

4. There is a lack of parallel between the mouse behavior and the model for the context-dependency shown in Figure 4. To directly validate the model presented in Figure 3, it seems more appropriate to conduct only VTA photostimulation in the same circular environment without MFB simulation.

5. It appears that the authors have rearranged the figures. Unfortunately, this has resulted in frequent incorrect figure references in the main text and some figure panels are missing, as detailed below.

a) Page 4, line 3, “also also”.

b) Figure 3j is missing.

c) Page 13, line 17, “Fig 5d-e” and “Fig. 4a” seem inadequate.

d) Page 14, line 2, “Fig. 4a”. seems wrong.

e) Page 14, line 18, “Fig. 3j” is missing.

f) Page 28, line 4, “(b)” should be “(c)”.

g) Page 31, line 20, “(h)” is wrong.

h) Page 33, line 5, “S3” is missing.

i) Page 42, I believe the second paragraph is irrelevant.

j) Page 80, line 1, "Figure 3" should be "Figure 4".

RESPONSE TO REVIEWERS' COMMENTS

Reviewer #1 (Remarks to the Author):

I am happy with the revision, and I recommend publication.

We thank the reviewer for accepting to review this new round, and for her/his very positive appreciation of our work.

Reviewer #2 (Remarks to the Author):

As an additional reviewer, my task is to evaluate the manuscript by Naudé et al., considering the feedback from other reviewers. While I acknowledge the comprehensiveness of the model presented, I concur with previous reviewers that it seems experimentally underdeveloped. The authors have mainly addressed prior comments by removing the original RL models and incorporating simulations derived from other studies, which I find reasonable but insufficient for fully addressing the raised concerns. Furthermore, I found the main text challenging to comprehend, requiring multiple readings to fully grasp the content. Thus, the readability could be significantly improved.

Overall, I am neutral about this manuscript at this stage. Although unifying the role of dopamine within a theoretical framework is appealing, the current version lacks important experiments and analyses necessary to better align the experimental data with the model. The manuscript could become suitable for publication once these issues are addressed.

We thank the reviewer for accepting to review our manuscript, to cross-comment previous reviews, and for her/his valuable inputs. Following the reviewer's suggestions, we (1) added new analyses on our experimental data to provide more validations of the model (new Fig. 6, page 39, and related Results page 15); (2) added new simulations of our previous experimental data on dopamine encoding (Fig. 2 h-i, page 32 and related Results page 10) and dopamine inhibition (new Supp. Fig. 4 page 83, and related Results page 15) to display more explicitly these features; (3) added a new figure to explain how the model predictions are tested by the experimental conditions (Fig. 4, page 36); and (4) generally reworked the text and figures to improve readability (all modifications are in red).

1. The experimental validations of the model's predictions seem insufficient. For instance, the manuscript does not elucidate how dopamine is released in their environment. After the initial learning phase (reinforcement), do dopamine neurons become active when mice approach the rewarded location more quickly? Does inhibition of dopamine neurons slow down the speed of mice after the reinforcement? Addressing some of these is important in strengthening the manuscript.

We understand and apologize that these two important features – dopamine correlating with speed and effects of dopamine inhibition – were not presented more

explicitly in the previous version. Indeed, our (recently published) article, using a related setup, i.e., electrical reinforcement of three locations in an open-field (Bousseyrol et al. 2023, Cell Rep.), showed that (1) dopamine neurons are more active when mice approach the rewarded location more quickly and (2) inhibition of dopamine neurons slows down the speed of mice after the reinforcement. Furthermore, as stated in the Introduction, “A large body of evidence supports that phasic DA neuron activity occurs just before self-paced movement initiation.” while “inhibition of DA neurons have been found to affect action initiation in some studies, in other settings manipulating DA activity did not have any effect on ongoing behavior”.

Nevertheless, dopamine encoding of speed does not differentiate between dopamine effects, as we now show in the manuscript; and conflicting effects of dopamine inhibition depend on the reward context, as we now precise.

First, our theory focuses on the effects of dopamine, not on how dopamine is released. Yet, we understand that this feature is important for understanding the simulations. We now display simulations of pre-movement dopamine release with different magnitudes, as observed in Bousseyrol et al. 2023. We explicitly show that correlations with speed are obtained if one of these hypotheses is considered : (1) pre-movement dopamine is a motor command affecting excitability (Fig. 2i) and (2) pre-movement dopamine reflects the history of previous rewards (Fig. 2h; as in most RL models). We added this material to highlight that this data together with standard analyses do not distinguish between models. We also reworked the discussion to point that our theory focuses on the effects of dopamine, not how dopamine is released, and is thus compatible with reward prediction errors (page 23).

Second, although the Bousseyrol data on dopamine inhibition was simulated in the previous version, it was grouped with other goal-directed data (e.g., lever-presses), and simulated with the reduced model. We now display the full MAGNet simulations (Supp. Fig. 4) in the different conditions (i.e. photo-inhibition of VTA dopamine neurons in reward and no-reward contexts) to further benchmark the model predictions. We thank the reviewer because these features are important and are now explained more explicitly.

2. Figure 5 addresses the conflicting results regarding the motivational role of dopamine. The authors mention in page 15 line 6, that “... as the effects of increased phasic DA signaling on the following movements depend on the animal distance to the goal ...”. It would be more convincing to directly test this hypothesis using their behavior data with VTA photostimulation.

We agree, and have added this new analysis in our new Fig. 6 (previous Fig. 5). We now directly demonstrate this distance-dependent effect in our experimental data with VTA photostimulation (Fig. 6b), and thank the reviewer for this suggestion.

3. The distinction between VTA photostimulation in Figure 1 and Figure 4 is unclear. In Figure 1, photostimulation appears to affect both synaptic plasticity (reinforcement) and excitability (motivation), whereas in Figure 4, photostimulation seems to specifically control excitability (motivation). What is the physiological basis for these assumptions?

We apologize, as we have indeed used a confusing framing in the previous version. For the sake of simplicity, we previously identified experimental conditions (e.g., reward context with VTA photo-stimulation) with model hypotheses (e.g., DA affecting both plasticity and excitability), because we considered that some DA effects would be prevalent in a given condition (e.g. after reinforcement, VTA photostimulation affects more excitability than plasticity, as dopamine is provided randomly). However, we understand, thanks to the reviewer's remarks, that this way of presenting the experimental data is confusing. We have now clarified this point, by separating model hypotheses (what kind of dopamine effect is considered, independent from experimental conditions), from expected dominant experimental effects (that depend on the experimental conditions), in both the text and figures. In particular, the new Figure 4 (see below), is devoted to clarify our model predictions.

4. There is a lack of parallel between the mouse behavior and the model for the context-dependency shown in Figure 4. To directly validate the model presented in Figure 3, it seems more appropriate to conduct only VTA photostimulation in the same circular environment without MFB simulation.

We respectfully disagree on this point, but also we apologize if the text related to Fig. 4 was misleading. Testing animals in extinction can lead them to re-explore other parts of the open-field quickly, in particular when dopamine is released at the periphery (because, as discussed above, VTA photostimulation affects both plasticity and excitability in all conditions). Hence, in Fig. 5 (previous Fig. 4), animals are tested with VTA photostimulation while the MFB reinforcement is still on. Consistently, in the model simulations in Fig. 3, the central location is still rewarded, even when there is additional dopamine release when the e-mouse is in the periphery.

5. It appears that the authors have rearranged the figures. Unfortunately, this has resulted in frequent incorrect figure references in the main text and some figure panels are missing, as detailed below.

We apologize for this. Indeed, as the reviewer noticed, we removed the previous comparison with RL models following the previous round of reviews. Unfortunately, we removed some panels with model predictions. In order to address the confusions pointed at comments 3-4-5, we now grouped model predictions with the experimental conditions for testing the predictions in the new Fig. 4. We thank the reviewer for pointing this out, we feel that this new figure and corresponding changes in the text now substantially improve the manuscript readability.

All the typos below have been corrected, we thank the reviewer for her/his thorough corrections.

a) Page 4, line 3, "also also".

b) Figure 3j is missing.

c) Page 13, line 17, "Fig 5d-e" and "Fig. 4a" seem inadequate.

- d) Page 14, line 2, "Fig. 4a". seems wrong.
- e) Page 14, line 18, "Fig. 3j" is missing.
- f) Page 28, line 4, "(b)" should be "(c)".
- g) Page 31, line 20, "(h)" is wrong.
- h) Page 33, line 5, "S3" is missing.
- i) Page 42, I believe the second paragraph is irrelevant.
- j) Page 80, line 1, "Figure 3" should be "Figure 4".

REVIEWERS' COMMENTS

Reviewer #2 (Remarks to the Author):

I appreciate the authors' efforts in addressing my concerns. The manuscript has been significantly improved, and I am now in support of its publication.

Reviewer #2 (Remarks to the Author):

I appreciate the authors' efforts in addressing my concerns. The manuscript has been significantly improved, and I am now in support of its publication.

We thank Reviewer #2 for having accepted to jump in during the review process and to cross-comment previous reviews.